# The AA7075–CS1018 Galvanic Couple under Evaporating Droplets

**Marvin Montoya** , **Juan Genesca** and **Rodrigo Montoya** *

Unidad de Investigación y Tecnología Aplicadas de la UNAM (UNITA), Facultad de Química,
Universidad Nacional Autónoma de México (UNAM), Apodaca 66628, Mexico
* Correspondence: rmontoyal@unam.mx

**Abstract:** The galvanic corrosion behavior of the AA7075–CS1018 couple was examined in dynamic electrolytes using the ZRA technique. A modified electrochemical setup was developed to support the use of thin-film gel and liquid electrolytes on metallic surfaces. This allowed the collection of chemical information, left behind by the liquid electrolyte during evaporation, through a thin-film gel. The analysis of the gel electrolyte film confirmed the acidification on AA7075 and the alkalinization on CS1018 but also offered novel insights on their dependence on the galvanic current. The galvanic current was proportional to the initial NaCl concentration in the range of 0.01 to 0.06 M. However, due to continuous evaporation, the NaCl concentration increased, limiting oxygen diffusion and decreasing the galvanic current, especially for electrolytes exceeding 0.06 M. The galvanic current was determined by considering the dynamic evolution (caused by the evaporation of the electrolyte film) of both the thickness of the electrolyte and its concentration.

**Keywords:** galvanic corrosion; AA7075–CS1018 couple; dynamic electrolytes; agar gel electrolyte; ZRA technique

## 1. Introduction

Atmospheric corrosion refers to the degradation of metallic materials that occurs when the atmospheric relative humidity surpasses the equilibrium relative humidity of any saturated solution on the surface of a metal [1]. During atmospheric corrosion the electrolyte can be an ultra-thin humidity film or an aqueous layer of hundreds of microns of thickness formed from aqueous condensation (or precipitation): such a thickness depends on different variables such as temperature, atmosphere composition, relative humidity, etc. [2,3]. A work reported by Tomashov [4] distinguished between various corrosion mechanisms depending on the moisture layer's thickness ($\delta$): dry atmospheric corrosion ($\delta$ < 10 nm), moist atmospheric corrosion ($\delta$ = 10 nm–1 $\mu$m), wet corrosion ($\delta$ = 1 $\mu$m–1 mm), and complete immersion ($\delta$ > 1 mm). While wet corrosion promotes oxygen diffusion through the electrolyte, increasing the cathodic reaction rate with a decreasing electrolyte layer thickness, dry corrosion (at low temperatures) occurs at a negligibly low rate. However, in order to avoid misunderstandings, throughout this work, the atmospheric corrosion term will be identified as the wet corrosion mentioned by Thomashov.

As a consequence, atmospheric corrosion is one of the most common types of corrosion responsible for the degradation of most outdoors metallic structures. As an electrochemical process, atmospheric corrosion involves simultaneous oxidation (oxidation of the metal) and reduction (oxygen reduction, in a basic and neutral medium) reactions, which allow for the dissolution (loss of material) of the metal. Weight loss is an indicator of the intensity of corrosion that can be easily determined in metals in which a uniform type of corrosion is favored. However, some metals spontaneously form a stable oxide film when exposed to outdoor environments, such as pure aluminum and aluminum alloys in which a local corrosion type (as pitting) is more common than generalized corrosion [5–8]. Corrosion

tests on Al alloys showed that there is no correlation between weight loss and maximum depth of pits in marine environments [9]; however, this method could be used to determine an average corrosion rate in aluminum alloys under specific conditions [10,11].

In the past few years, many researchers have studied aluminum alloy's behavior under atmospheric corrosion [12–16]. Most of these studies were performed under highly contaminated (mainly $Cl^-$, $SO_2$) atmospheres, together with a high relative humidity [17,18].

On the other hand, as is well known, Al alloys are normally assembled with other metals (or conductive materials) to obtain specific desirable properties, creating a galvanic couple and leading to the anodic dissolution of the metal with lesser nobility. Several authors have studied such a phenomenon in Al (and its alloys) when coupled with different materials such as copper [19], carbon fiber-reinforced polymers [20,21], stainless steel [22–24], or zinc [25]. Based on previous research, it is understood that various factors, including but not limited to temperature, solution concentration, pH value, standard reduction potential of metals, and geometric factors such as the distance between the coupled materials, influence galvanic corrosion behaviors. Moreover, and more specifically, atmospheric corrosion is characterized by limited electrolyte volumes, differential aeration, and salt concentration parameters [26,27]. In fact, it has been found that the presence of hygroscopic salts will invariably prolong the formation of electrolytes on metallic surfaces, leading to an extended period of corrosion. Various studies have investigated the phenomenon of corrosion in evaporating droplets [27–30]. The study by Tsutsumiet et al. [30] found that the likelihood of pitting corrosion occurring and the size of the pits decrease as the diameter and thickness of the droplet decrease. They also noted that the rate of oxygen diffusion in the liquid is higher for drops with heights of less than 20 μm. Nishikata et al. [31] investigated the influence of electrolyte layer thickness and pH on the initial stage of atmospheric corrosion. They found that an electrolyte thickness of 20 to 30 μm results in the highest corrosion rate. In addition, the pH of the electrolyte does not have a significant impact on the atmospheric corrosion rate, except in situations where the electrolyte layer is thick (>1 mm). In a study by Tang et al. [32], it was observed that increasing the volume of the electrolyte leads to an enlarged cathode area and a higher anodic current density. This, in turn, accelerates the initiation and propagation of atmospheric corrosion.

Several researchers have focused their attention on micro-droplet condensation/ evaporation phenomena, which are closely linked to atmospheric corrosion. Investigations into micro-droplets have been approached from various perspectives, such as by examining the effects of deliquescence of soluble salt particles as a cause for the acceleration of atmospheric corrosion [33] as well as the impact of electrochemical polarization on micro-droplet formation [34]. Such studies have revealed that cathodic polarization accelerates micro-droplet formation regardless of metal type, electrolyte species, composition of air, and environmental humidity. Additionally, Tang et al. [35] developed a mechanical model for micro-droplet formation consisting of three steps: (i) horizontal movement from the primary droplet along the metal surface, (ii) evaporation from the primary droplet and recondensation on the metal surface, and (iii) condensation of moisture from the air. Bian et al. [36], on the other hand, studied the change in the coverage range of micro-droplets on metal surfaces during early atmospheric corrosion through optical microscopy. More recently, a study predicting the effect of droplet geometry and size distributions on atmospheric corrosion, using finite element analysis, and a review on modeling corrosion under droplet electrolytes to predict atmospheric corrosion rates have been published [37,38]. Both of these studies highlight the need to consider the combined impact of corrosion and the changing geometry of unstable droplets (due to condensation/evaporation) for comprehensive atmospheric studies.

Two opposing phenomena that influence oxygen reduction during electrolyte film drying are improved oxygen access to the electrode surface due to reduced electrolyte layer thickness and the decrease in oxygen solubility with increasing salt concentrations [39]. Additionally, the total limiting oxygen reduction current as a function of the evaporation time depends on the electrolyte's geometry, with planar droplet geometry leading to an

increase in current and hemispherical and flattened hemispherical cap droplet geometry leading to a decrease in current [40].

Some studies have investigated the galvanic atmospheric corrosion between aluminum alloys (AAs) and carbon steel (CS) using liquid electrolytes [41,42]. Generally, an active behavior of AAs has been observed with variations in the thickness of the liquid electrolyte. In contrast, our research group conducted a series of studies utilizing hydrogels as electrolytes for the AA7075–CS1018 galvanic couple [43–45]. These studies were complemented by finite element analysis (FEA), obtaining the distributions of potential, current, and pH in static electrolytes. Nevertheless, during atmospheric corrosion, droplets tend to evaporate or condense due to changes in the atmosphere, resulting in the formation of dynamic electrolytes. As a result, corrosion kinetics are primarily determined by the electrochemical reactions occurring in these dynamic electrolytes. Therefore, the objective of this study was to investigate the galvanic corrosion of AA7075 coupled with CS1018 carbon steel under dynamic NaCl electrolytes at an initial neutral pH. Simultaneously, two types of electrolytes, namely, a liquid electrolyte (the droplet) and a thin agar gel film (the base of the droplet), were employed in order to generate a witness with the chemical information that the droplet leaves once it evaporates.

## 2. Materials and Methods

### 2.1. Galvanic Couple Specimens

AA7075-T6 (AA7075) and AISI1018 (CS1018) were used to form a galvanic couple. The nominal chemical (in wt%) compositions of both materials are given in Tables 1 and 2, respectively. The electrodes were prepared by cutting AA7075 and CS1018 into 12 mm × 25 mm × 2 mm samples. The surfaces of both specimens were mechanically polished using different SiC papers up to 800 grit. Following the polishing steps, the samples were cleaned with distilled water and subsequently washed with acetone before placing the electrolytes films (under evaporation conditions) on the metallic surface.

**Table 1.** The chemical composition of CS1018.

| Element | C | Si | Mn | Cu | Cr | Ni | Mo | N | P | S | Al | Sn | V | Fe |
|---|---|---|---|---|---|---|---|---|---|---|---|---|---|---|
| wt% | 0.18 | 0.12 | 0.65 | 0.07 | 0.03 | 0.031 | 0.007 | 0.0067 | 0.005 | 0.005 | 0.005 | 0.005 | 0.001 | Bal. |

**Table 2.** The chemical composition of AA7075-T6.

| Element | Zn | Mg | Cu | Si | Fe | Cr | Mn | Al |
|---|---|---|---|---|---|---|---|---|
| wt% | 6.67 | 2.89 | 0.91 | 0.89 | 0.35 | 0.33 | 0.13 | Bal. |

### 2.2. Preparation of the Electrolytes

The experiments were conducted under a biphasic electrolyte formed by a liquid-phase electrolyte and a gel-phase electrolyte. The pH of both phases was adjusted to 7.0 using a dilute NaOH solution. The liquid electrolytes were prepared using NaCl solutions of 0.01, 0.06, 0.1, 0.3, and 0.6 M. Meanwhile, the gel electrolytes used in this study were prepared from laboratory-grade powder agar, which was dissolved in NaCl 0.01, 0.06, 0.1, 0.3, and 0.6 M solutions. The solution was stirred at 70 °C and homogenized. Before the electrolyte acquired a gel consistency, it was poured between two parallel plates, as reported by Montoya et al. [43]. Finally, the upper plate was pressed down to the desired electrolyte thickness. The electrolytes' final thickness was about 250 ± 20 μm after they had cooled down to room temperature. The use of a gel electrolyte in this investigation is based on the fact that hydrogels, as agar and agarose, preserve the properties of diffusion coefficients and do not affect the reversible charge transfer in electrode reactions [43,45,46].

### 2.3. Experimental Setup of the Dynamic Droplet Process

The AA7075 and CS1018 plates were arranged horizontally, side by side, with a separation of 0.5 mm, as depicted in Figure 1a. Subsequently, the agar gel electrolyte was applied onto the galvanic couple, as illustrated in Figure 1b. The galvanic coupling current was then recorded using the ZRA technique. Approximately 120 s after initiating the ZRA measurement, a droplet of NaCl with an initial volume of 200 μL was deposited onto the horizontal surface of the agar gel electrolyte, as shown in Figure 1c. The agar gel electrolyte was used to capture the overall chemical information produced and left by the electrolyte droplet. Indeed, once the complete evaporation of the liquid electrolyte had taken place, and prior to the evaporation of moisture from the agar gel electrolyte, the gel was removed from the galvanic couple, and the pH of the gel above the AA and the CS was measured. Both the top and front views of the experimental setup can be seen in Figure 1d,e. Petroleum jelly (Vaseline®) was used around the electrolytes to prevent any leakage or spillage of the liquid electrolyte onto the two metals' surfaces.

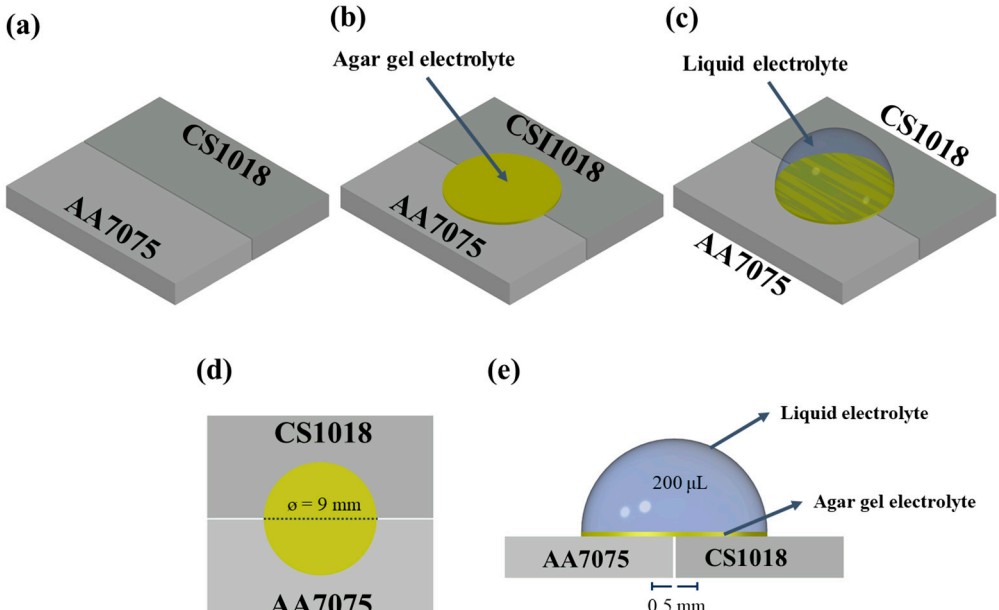

**Figure 1.** (**a**–**c**) Experimental setup for measuring the corrosion of the AA7075–AISI1018 galvanic couple exposed to dynamic electrolytes (in continuous evaporation). (**d**) Top and (**e**) front view of the experimental setup.

The zero-resistance ammeter (ZRA) test was conducted to monitor the galvanic corrosion current between the AA7075 and CS1018 samples. The cathode sample and anode sample had an area ratio of 1:1, with a total exposure area of 2.54 cm$^2$, in a circular shape. In the experimental setup, the reference electrode (REF) terminal of the potentiostat was connected to the working electrode (WE), as shown in Figure 2 [11]. During the galvanic corrosion test, the anodic current was measured at AA7075 relative to CS, which acted as the cathode. The ZRA test was conducted during the total evaporation time of the droplet electrolyte using a potentiostat (Bio-logic SP-300, Claix, France). Simultaneously, the weight loss of the electrolytes was measured using an analytical balance (OHAUS Scout, H-7293, Parsippany, NJ, USA). The general conditions of the chamber were 25 °C and 50% absolute humidity. All the electrochemical measurements were performed at least three times.

Identical experimental conditions were maintained for a subsequent set of trials, omitting electrochemical measurements while incorporating pH assessments in relation to the initial droplet concentration. The final pH of each electrode was determined using a pH meter (LAQUAtwin pH-22 device, HORIBA Scientific, Edison, Kioto, Japan).

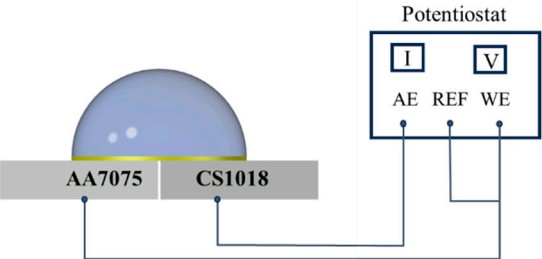

**Figure 2.** The electrical connection between the electrochemical cell and the potentiostat in the ZRA configuration is used to measure the galvanic current between the AA7075 and AISI1018 couple. The AA7075 specimen serves as the working electrode, while AISI1018 acts as the counter electrode. However, due to the design of the electrochemical cell, a third electrode (reference electrode) has not been utilized.

## 3. Results and Discussion

The evaporation of electrolytes was assessed by monitoring the gradual weight reduction over time following the deposition of a liquid electrolyte droplet onto the surface of the agar gel electrolyte. Figure 3a shows the cumulative mass losses measured during the evaporation of the electrolytes (both droplet and humidity into the gel electrolyte) as a function of the time, exposed to the atmospheric conditions mentioned in Section 2.3. The average evaporation rates, represented by the slopes of the evaporation curves at different NaCl concentrations (dotted lines), are presented in the inset of Figure 3a. Generally, an increase in the NaCl concentration in the solution resulted in a decrease in cumulative evaporation. Similar findings have been reported by several authors [47–49]. According to previous research, factors such as the increase in NaCl concentration and the precipitation of salts (hygroscopic salts) on the surface are responsible for the decrease in the rate of electrolyte evaporation. These factors can explain the behavior observed during our experiment in the later stages of evaporation, where the linear fit was not adjusted, as shown in Figure 3a. However, Orlova et al. [50] reported that the heat of vaporization increases with increasing salt concentrations in a water solution, resulting in an increased amount of heat needed for evaporation. Therefore, the decrease in the evaporation rate (inset of Figure 3a) of the drop electrolyte during the initial hours of evaporation was also attributed to changes in the enthalpy of vaporization caused by differences in the volume of water throughout the evaporation process.

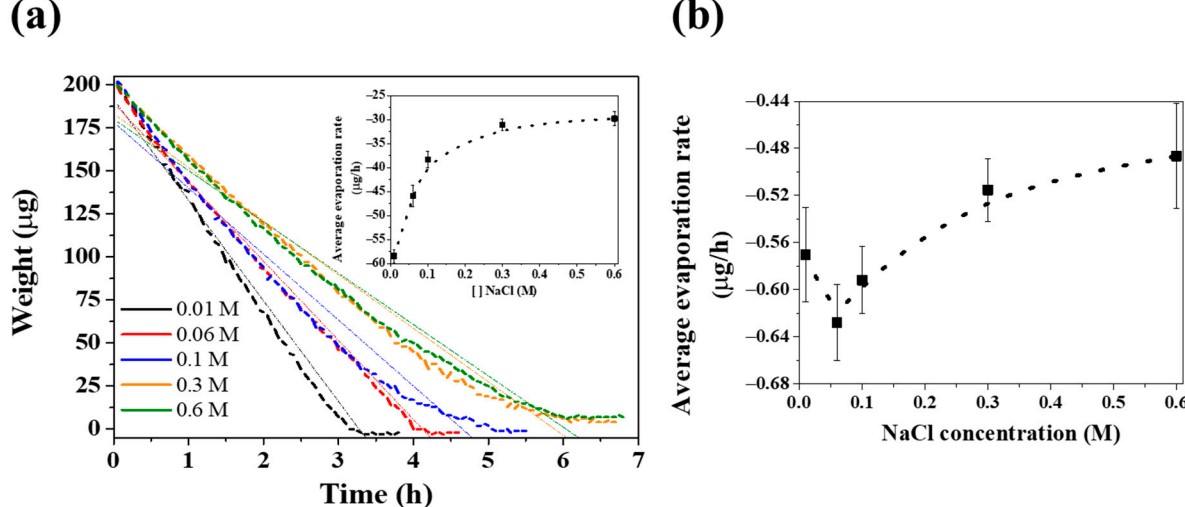

**Figure 3.** (**a**) Measurement of the weight loss of humidity in the droplet + gel electrolyte (the inset illustrates the average evaporation rates in µg/h units) and (**b**) evaporation rates in µg/h units of humidity of the gel electrolyte.

On the other hand, the average evaporation rates of the humidity of the gel electrolyte are presented in Figure 3b. As anticipated, the rate of evaporation decreased with the rising concentration of NaCl. However, this behavior was observed specifically in concentrated gel electrolytes (0.1–0.6 M). This behavior could not have been related to an increase in the enthalpy of vaporization because this phenomenon applies only to liquid electrolytes. Consequently, a higher electrolyte concentration can influence conductivity and ion mobility, thereby enhancing the conductivity and increasing the viscosity of the electrolyte [51]. A higher viscosity means that the solution is more resistant to flow, including the evaporation process. However, for dilute gel electrolytes ranging from 0.01 M to 0.1 M, the trend is reversed. This behavior could be attributed to the surplus of freely available water molecules, characterized by an enhanced mobility, enabling them to readily move away from the gel matrix towards the evaporation front.

Based on the evaporation rate over time, the global evaporation process can be divided into three stages, as reported by Cui et al. [52]. The first stage corresponds to a constant evaporation rate, while the second stage shows a decreasing rate of evaporation with a nonlinear behavior. The final stage is the residual evaporation stage, where the humidity of the gel electrolyte undergoes evaporation. In the first stage, the electrolyte contains an excess of free water, and, therefore, the evaporation process is controlled by the relative humidity of the atmospheric conditions. In our experiment, the evaporation rate was found to be similar for concentrations from 0.06 to 0.1 M ($-50$ µg/h) and from 0.3 to 0.6 M ($-40$ µg/h). Moving to the second stage, the rate of evaporation decreases over time as the excess surface water has evaporated, resulting in a distribution of the salt concentration. This distribution is characterized by higher concentrations at the droplet's surface, where water evaporates rapidly, leaving behind an accumulation of salt. Conversely, the central region of the droplet experiences slower evaporation rates, resulting in lower salt concentrations [53]. The high salt concentration on the droplet surface may impede the evaporation rate by inhibiting the transference of water molecules from the droplet to the external environment. As evaporation progresses, the precipitated salt begins to control the weight loss process by absorbing moisture from the environment. Lastly, in the third stage, most of the free NaCl electrolyte, along with the humidity of the gel electrolyte, has evaporated.

ZRA tests were performed on a couple of galvanic cells in order to measure and compare the galvanic coupled current between different NaCl solutions. Figure 4 presents the time-dependent galvanic current flowing in the AA7075–CS1018 couple obtained as a result of the electrolyte evaporation process (dynamic electrolytes). Additionally, examples of how the gel electrolyte and liquid–gel electrolyte evolved over time during the evaporation process are shown at the bottom of the current signals, respectively. The set of curves in Figure 4a demonstrates that the galvanic current is affected by the NaCl concentration. It is important to note that the diameter of the droplet base remained constant during the evaporation process. The positive current density values registered indicate that AA7075 is corroding in all cases, while CS1018 acts as the cathode.

A general trend is observed in the galvanic current over time for dilute electrolytes (0.01, 0.06, and 0.1 M). Initially, there is a decrease in galvanic current in the first few hours (approx. 3 h) after the liquid electrolyte is added on the gel electrolyte. This is followed by an increase in galvanic current, until it reaches a maximum value. Finally, a decrease in galvanic current is detected, eventually approaching zero. The galvanic current reaches a maximum value before the almost complete evaporation of the 200 µL original electrolyte droplet, indicating that the rate of evaporation limits the increase in galvanic current. Indeed, following the evaporation of the droplet, the over-saturated humidity film remaining on top of the gel electrolyte hinders both the mass transportation and diffusion of oxygen. This difficulty results in a decrease in the intensity of the cathodic reaction and the overall galvanic current. This observation aligns with prior studies that document a maximum current during evaporation within a few dozen microns of electrolyte thickness [54].

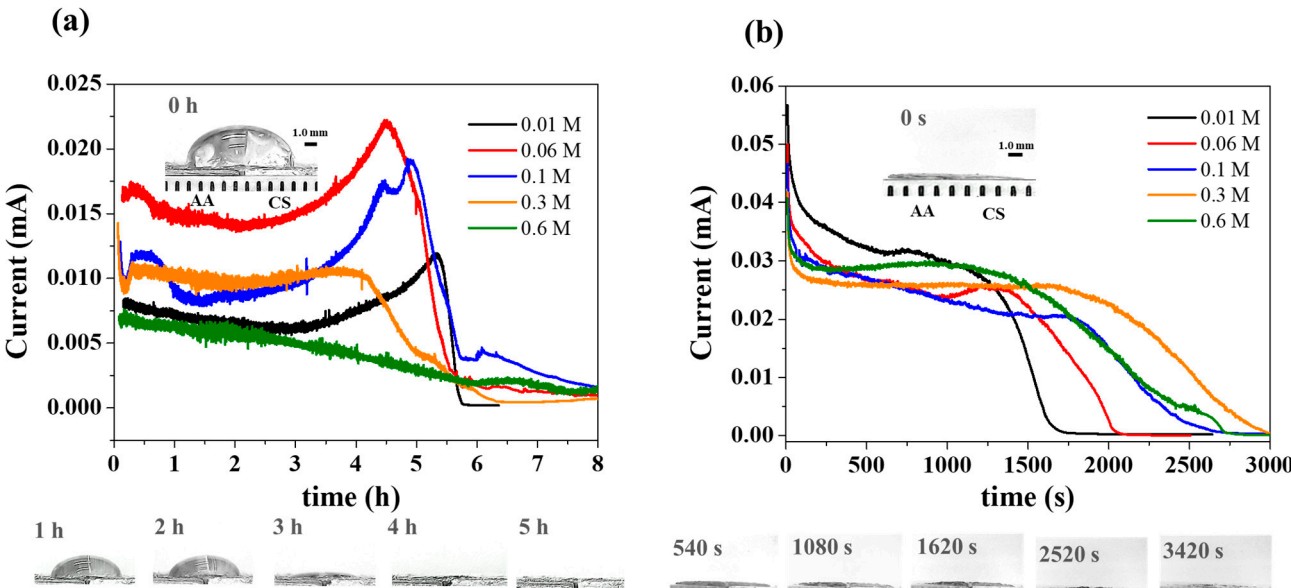

**Figure 4.** Time-dependent plot of the galvanic current flowing between AA7075 and CS1018 under a (**a**) liquid–gel electrolyte for different NaCl concentration and a (**b**) gel electrolyte. The inset in (**a**) displays a real photograph of the liquid–gel electrolyte on the metals without evaporation (0 h). Below the signal, it demonstrates that increasing the time of exposure from 1 h to 5 h results in a decrease in the volume of the liquid electrolyte. Similarly, (**b**) illustrates the evaporation of moisture from the gel electrolyte as a function of time (s).

As illustrated in Figure 4b, it is essential for the reader to note that such an effect (the presence of a maximum current) is not observed during the loss of humidity from the gel electrolyte without prior droplet evaporation, even if the NaCl concentration is increased. This is due to the absence of NaCl build-up from the droplet through evaporation. These factors lead to the consistent behavior of the AA7075 surface [55]. On the other hand, as the free water content decreases, it promotes the de-solvation and uniform deposition of ions, which also enhances ionic conductivity [56]. As a result, a higher galvanic current was observed when the gel electrolyte was used (Figure 4b) in comparison to when liquid–gel electrolytes were used (Figure 4a).

The results above are also consistent with the findings reported by Tomashov [4] regarding the relationship between the atmospheric corrosion rate and the amount of moisture on the metal surface. During the early stages of the evaporation process, the corrosion rate of the AA7075–CS1018 couple is largely unaffected, with only a slight decrease in galvanic current due to a reduction in the rate of oxygen diffusion through the thicker film. This is attributed to the constant diffusion layer that has already formed, which corresponds to thicker electrolyte layers. However, as the moisture content (or thickness) decreases, a layer of solid electrolyte forms on the surface of the AA7075–CS1018 couple, containing excess moisture and a high concentration of NaCl. This allows for the maximum galvanic current to flow. Eventually, the corrosion process transitions from purely electrochemical to chemical oxidation, causing a decrease and deceleration in the galvanic current's density. In these later stages, the dry solid electrolyte cannot be considered as a continuous film with the properties of an electrolyte. As a result, the galvanic current approaches zero due to the high resistance of the very thin layers, obstructing the ionization and dissolution reactions of AA7075.

The evolution of the galvanic current for electrolytes at 0.3 and 0.6 M demonstrated a different pattern of behavior when compared to the galvanic current behavior of diluted electrolytes (Figure 4). According to the evaporation behavior reported in Figure 3, an almost total weight loss of the electrolyte at 0.3 and 0.6 M occurs around 6 h, which corresponds to a minimum current density value. Previous research by Dubuisson et al. [57]

demonstrated that thinner electrolytes (δ > 800 μm) increase the cathodic reaction as they allow gasses to diffuse to the surface more rapidly, resulting in a diffusion-controlled process. Our findings indicate that an increase in the initial NaCl concentration from 0.01 to 0.06 M resulted in an increase in galvanic current between AA7075 and CS. However, the galvanic current decreased as the initial NaCl concentration further increased to 0.1 M. This suggests that there is a critical threshold of NaCl concentration beyond which the galvanic current does not proportionally increase.

Figure 5a schematically shows the evolution of NaCl concentration and oxygen diffusion as a function of the evaporation process in initially diluted NaCl electrolytes, while Figure 5b depicts the same for high-NaCl-concentration electrolytes. The larger the number of upward arrows, the higher the concentration of both NaCl and $O_2$. In diluted electrolytes, there is an increase in the number of aggressive ions attacking the material during the evaporation process, as well as a greater diffusion of oxygen, as indicated by the upward arrows in the diagram (Figure 5a). As the NaCl concentration continues to rise by evaporation, diffusion slows down, and higher NaCl concentrations impede the diffusion of oxygen during evaporation (Figure 5b). Chung et al. [58] also observed an analogous behavior, noting that a rise in aggressive ions within the electrolyte leads to two outcomes. Firstly, an accelerated rate of anodic dissolution occurs as a larger number of aggressive ions target the material. Secondly, there is a decline in the rate of anodic dissolution due to a diminished availability of free water molecules required to solvate the metal ions generated during corrosion. The statement above can initially explain why the corrosion process is primarily controlled by the NaCl content in the electrolytes when NaCl concentrations exceed 0.06 M. In fact, reducing the thickness of the electrolyte layer enhances the access of oxygen to the electrode surface, resulting in an increase in the galvanic current from a NaCl concentration of 0.01 to 0.06 M. This occurs because a thinner electrolyte layer allows for the easier diffusion of oxygen towards the electrode, thereby facilitating the reduction process. On the other hand, when the initial concentration of NaCl increases to 0.6 M and continues to rise during the evaporation process, there is an instant decrease in the solubility of oxygen, as shown schematically in Figure 5b. This means that, with higher NaCl concentrations, less oxygen is available in the electrolyte solution, and, consequently, less oxygen can participate in the reduction process [39]. These two opposing effects highlight the complex interplay between electrolyte properties and oxygen reduction kinetics during the evaporation of an electrolyte film. The balance between these phenomena ultimately determines the overall efficiency of the oxygen reduction reaction in this system.

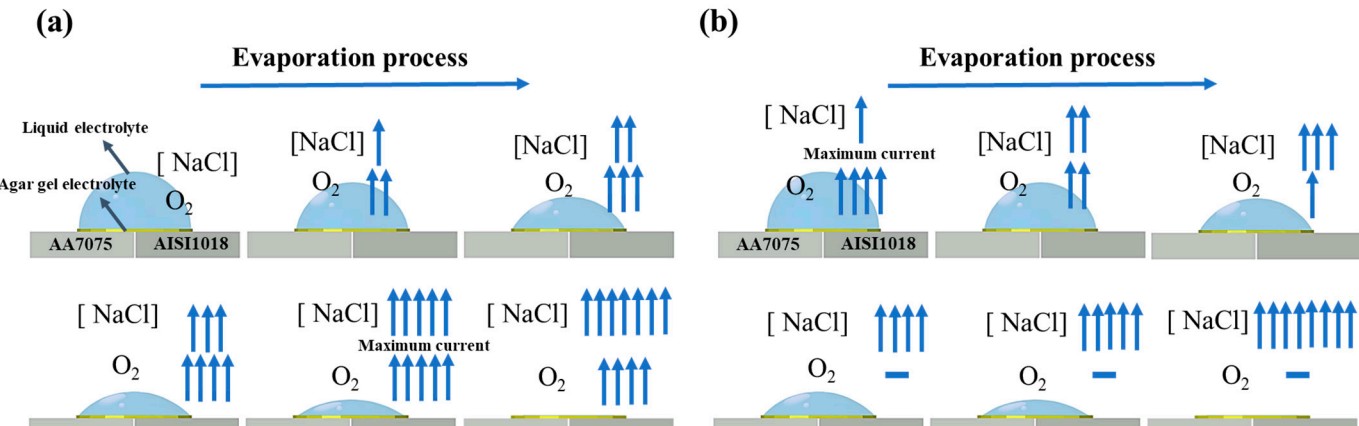

**Figure 5.** Schematically evolution of NaCl concentration and oxygen diffusion as a function of the evaporation process in electrolytes with a (**a**) low initial NaCl concentration and (**b**) a high initial NaCl concentration. The upward arrows in the diagram indicate an increasing concentration of NaCl. More arrows represent a higher concentration of these species.

Figure 6 shows the initial (without evaporation), maximum, and average galvanic currents calculated for different NaCl solutions. The maximum current, also known as the diffusion-limited current, depends not only on the thickness of the electrolyte but also on the concentration of oxygen in the electrolyte, which, as explained before, is influenced by the salt concentration. Indeed, in increase in salt concentration in the electrolyte leads to a decrease in the solubility of oxygen, resulting in a decrease in the galvanic current produced by the cell [58]. This is because the salt concentration is lower in the central region of the droplet and higher on the droplet's surface region [53], thus limiting the diffusion of oxygen. Figure 6a illustrates the initial, maximum, and average galvanic currents of both the liquid electrolyte droplet and the humidity contained into the gel, while Figure 6b demonstrates the current parameters during the evaporation of the humidity contained into the agar gel electrolyte. The initial current is determined as the average current during the first few minutes of current registration. This parameter helps measure the impact of the initial concentration on the galvanic current when no evaporation processes occur. In the case of the evaporation of the humidity contained into the agar gel electrolyte, the initial current decreases as the initial NaCl concentration increases. However, an interesting observation is seen in Figure 6a where the trend of the initial current shows a maximum value at 0.06 M. In fact, as the concentration of NaCl increases, there are more $Na^+$ and $Cl^-$ ions available to carry the current, which leads to an increase in the initial current. However, at a certain concentration of NaCl, the liquid electrolyte becomes ionic-saturated. This saturation primarily occurs in the agar electrolyte, which limits the transportation of mass and oxygen on AA7075 and CS1018. As a result, the dissolution of AA7075 is slower. Furthermore, Figure 6b also presents the trend of average currents over time for different NaCl concentrations, which decrease as the NaCl concentration increases. It is worth noting that the galvanic current in solid electrolytes is higher compared to that of both kinds of liquid electrolytes. This is most probably due to the better access of oxygen to the metal through the solid electrolyte [39] and the absence of over-saturation from the liquid electrolyte drops. The results obtained from these current parameters are consistent with the results presented in Figures 4 and 5.

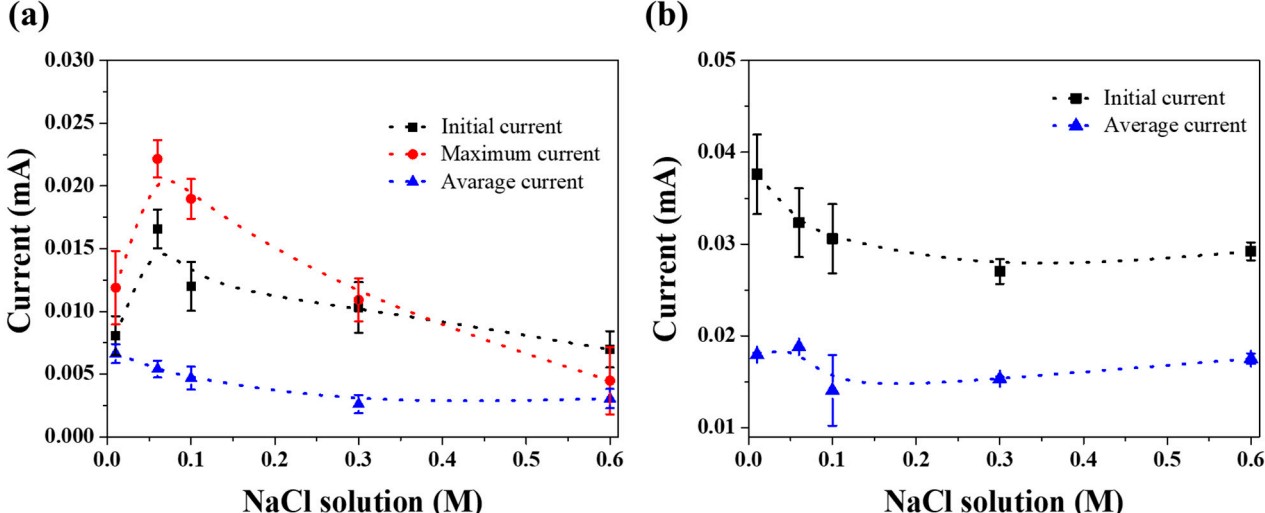

**Figure 6.** Trend of initial (without evaporation), maximum, and average galvanic current at different NaCl solutions where AA7075 (WE1) serves as the working electrode in (**a**) a liquid agar electrolyte and (**b**) an agar electrolyte.

A method to quantify the effect of NaCl concentration on corrosion behavior is through a rapid statistical analysis of the fluctuation in the galvanic current signal. The fluctuation in the galvanic current between AA7075 and CS1018 was relatively short, with a high frequency, which could be attributed to the highly unstable electrochemical process during

the evaporation process by the dissolution of AA7075 (and/or CS1018). To predict the type of corrosion present on the galvanic couple (localized or uniform or both), a localization index (LI) estimation was derived using Equation (1) [59,60].

$$LI = \frac{\sigma_i}{i_{rms}} \tag{1}$$

The standard deviation of the galvanic current's density, $\sigma_i$, and the root mean square (RMS) value of the galvanic current obtained using the ZRA technique, $i_{rms}$, are summarized in Table 3 for the AA7075–CS1018 couple under different testing conditions. $\sigma_i$ estimation increase from 0.01 to 0.06 M but decrease as the NaCl concentration increases. This trend in $\sigma_i$ behavior aligns with the galvanic current shown in Figure 4. These results have been consistently associated with a higher corrosion rate, as reported by Mansfeld and Xiao [61]. Furthermore, higher $\sigma_i$ values are also related to increased fluctuations in current, which are linked to the formation of metastable pits and higher frequencies of pit nucleation [62]. The LI estimations (Table 3) fall in the range from 0.1 to 1.0, which is typically associated with localized corrosion [63]. Indeed, Maeir et al. [64] reported that, as the rates of evaporation and salt concentration increase, the pitting rate significantly decreases due to the low solubility of oxygen in an aqueous solution on an Al-Mg-Si alloy. Therefore, based on both the $\sigma_i$ and LI values presented in Table 3, it is possible to establish a threshold for the NaCl concentration beyond which a change in the corrosion mechanism and kinetics occurs in the AA7075–CS1018 galvanic couple.

**Table 3.** Localization index estimated from ZRA data in the AA7075–CS1018 galvanic couple.

| NaCl Solution (M) | 0.01 | 0.06 | 0.1 | 0.3 | 0.6 |
|---|---|---|---|---|---|
| $\sigma_i$ (mA) | 0.002671 | 0.007226 | 0.005351 | 0.004087 | 0.001591 |
| $i_{rms}$ (mA) | 0.007163 | 0.009023 | 0.007112 | 0.005129 | 0.003440 |
| LI | 0.373 | 0.801 | 0.752 | 0.797 | 0.463 |

Although the LI estimations for the NaCl solutions indicate localized corrosion on the AA7075 anode metal, the LI cannot be solely relied upon as a definitive indicator of a corrosion mechanism due to its statistical nature. Visual analysis of the current signals depicted in Figure 4 reveals a lack of low-frequency and high-intensity fluctuations typically associated with localized corrosion. This inconsistency between the IL estimates and the current signal profiles may be attributed to the galvanic coupling between AA7075 and CS1018. While AA corrosion typically occurs as localized corrosion in depassivating conditions, when in contact with CS in a short circuit, it acts as a sacrificial anode leading to different active zones on the metal surface. Additionally, the current signal may be affected by the potential self-corrosion of CS. Despite this, the IL calculation, being highly sensitive, identified multiple high-intensity fluctuations linked to localized corrosion, resulting in high IL values. To observe the impact of various NaCl solutions on the surface of the AA7075–CS1018 galvanic couple, the early stages of evaporation were continuously monitored during our experiment, as shown in Figure 7.

Figure 7 illustrates the corrosion behavior of an AA7075–CS1018 galvanic couple in different NaCl solutions. The exposure of the AA7075–CS1018 couple to various NaCl solutions results in a notable generation of bubbles located beneath the gel electrolyte. These bubbles can be associated with the reduction of $H^-$ to produce $H_2$ due to the increase in cathodic current in AA7075 caused by the self-corrosion of CS1018 [41,65] and/or the negative difference effect present in aluminum and magnesium alloys [66–68]. Additionally, the solution can directly contact the aluminum by rupturing the passive film, leading to hydrogen evolution [69]. On the other hand, self-corrosion is mainly observed at a NaCl concentration of 0.06 M, as depicted in Figure 7. Conversely, as the NaCl concentration increases, $H_2$ evolution in the AA7075 decreases, aligning with the galvanic current's behavior, as shown in Figure 4.

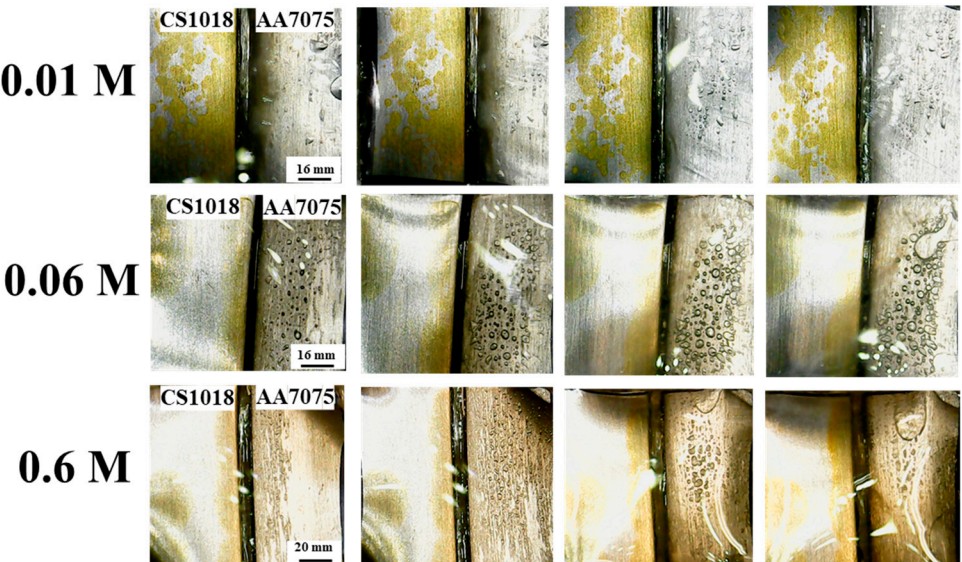

**Figure 7.** Optical microscopy top-view images of the corrosion behavior on the AA7075–CS1018 galvanic couple in a NaCl solution at 0.01, 0.06, and 0.6 M. These images provide evidence of $H_2$ evolution on the surface of AA7075 as well as the occurrence of self-corrosion on the CS1018 component, primarily in dilute electrolytes.

The autocatalytic dissolution process on the CS1018 side [41,70–73] primarily occurs in diluted solutions, resulting in the accelerated corrosion of AA7075, as shown in Figure 4. Previous research has demonstrated that, when AAs are immersed in NaCl solutions, the presence of Fe species results in the formation of a less efficient protective film consisting of $Al(OH)_3$ and $Fe(OH)_3$, which significantly reduces the corrosion resistance of AAs [73]. As a result, a higher concentration of Fe chemical species generated by the self-corrosion of CS can migrate and deposit on the AA side, further accelerating its dissolution.

It is significant to mention that the different changes in the present signal acquired through ZRA are a result of the multiple anodic and cathodic reactions taking place on the surface of the electrodes. The reason for the variation observed in the IL estimation and galvanic current profiles analysis could be attributed to the overlap of the characteristic signal of self-corrosion in CS1018 (uniform corrosion) onto AA7075's current signal (total current associated with galvanic couple corrosion). According to Laurent et al. [69], the overall effect at the active site is a current amplifier. The findings mentioned above align with the total current behavior observed in Figure 4, where the NaCl concentration with more $H_2$ evolution (0.06 M) presents a greater galvanic coupling current.

On the other hand, according to Kadowaki et al. [41,65], the total of both the internal anodic and cathodic currents should be equal for the entire galvanic couple. Based on this, a decrease in NaCl concentration results in an imbalance between these anodic and cathodic currents, which leads to the presence of an anodic current component in the metal which initially acts as the cathode in the electrochemical reaction. Consequently, a small increase in the NaCl concentration, from 0.01 to 0.06 M, allows for a reduction in the anodic current in the CS1018 metal. This reduction is compensated for by an increase in the anodic current of the AA7075 cathode. In fact, using finite element method (FEM) simulations, the above-mentioned researchers discovered that the number of pits (anodic sites) on AA6016 decreased as the NaCl concentration decreased. As a result, the anodic current on AA6016 also decreased. Furthermore, the cathodic current observed on the low carbon steel surface remained consistent, regardless of the NaCl concentration, within the range of less than 5% by weight. The present investigation has revealed that the anodic current in CS1018 could have been affected by the initial NaCl concentration and the evaporation of the electrolytes. This finding is backed by the analysis of electrode corrosion. The observed rise and subsequent decline in self-corrosion on the CS1018 surface,

resulting from the increase in NaCl concentration, implies a shift in the balance between the anodic/cathodic currents of CS1018 and AA7075, as previously noted by Kadowaki et al. Moreover, Tahara and Kodama [74] reported that, for the Zn/Fe galvanic couple, the transition zone (change from Zn potential to Fe potential) increases monotonously by increasing the NaCl concentration. Consequently, the sacrificial effect of Al reaches much farther in concentrated-NaCl electrolytes than in low-NaCl electrolytes, due to the low resistivity of electrolytes with high NaCl concentrations.

According to the previous results, the corrosion mechanism of the AA7075–CS1018 galvanic couple under the influence of dynamic electrolytes can be explained as follows: a significant portion, but not the majority, of the cathodic current that decrease in CS1018 is compensated in AA7075 at a NaCl concentration of 0.01 M. As a result, AA7075 exhibits a similar cathodic behavior, which leads to an increase in the anodic current in CS1018. Consequently, strong self-corrosion occurs in CS1018 at the lowest NaCl concentrations. The anodic current in CS1018 decreases as the NaCl concentration increases from 0.01 M to 0.06 M, as previously observed by Kadowaki and Felde et al. [75], while simultaneously increasing until reaching a maximum value in AA7075 (Figure 6), accompanied by an increase in the cathodic current in CS1018. This transition in corrosion behavior signifies a shift in the galvanic couple. Subsequently, at a NaCl concentration of 0.06 M, the high anodic current facilitates a significant dissolution on the AA surface. This dissolution may impact the transportation of Fe species to the AA7075 side due to the continuous movement of the gel electrolyte by $H_2$ bubbles due AA7075 dissolution. As the NaCl concentration increases from 0.06 M to 0.6 M, the galvanic current decreases in a manner corresponding to the decline in the evolution of $H_2$.

Figure 8 illustrates the results of the pH measurement in relation to the NaCl concentration. The pH changes resulting from cathodic reduction and anodic dissolution are measured on the gel electrolyte after evaporating the liquid electrolyte (200 µL). The alkaline shifts observed over the CS1018 cathode range from a pH of 10.4 to 10.8, while the shifts over the AA7075 anode range from a pH of 4.0 to 4.6. This indicates that corrosion takes place on both the passive film and the aluminum matrix, with both $H^+$ and $H_2O$ being reduced simultaneously [76]. The anodic dissolution of Al leads to the formation of hydrated $Al^{3+}$ and its hydroxy complexes (Equation (2)). As the concentration of $Al^{3+}$ increases, the pH value decreases due to the hydrolysis process [77]. Due to the high acidity of $Al^{3+}$, it rapidly reacts with water, lowering the pH of the electrolyte (Equation (4)) [78].

$$Al \rightarrow Al^{3+} + 3e^- \tag{2}$$

Oxygen reduction on the CS1018 surface is as follows:

$$O_2 + 2H_2O + 4e^- \rightarrow 4OH^- \tag{3}$$

While the pH remains consistently acidic on AA7075 and alkaline on CS1018 after droplet evaporation, the pH values for AA7075 show a slight increase with the increase in NaCl concentration from 0.01 to 0.1 M, as shown in Figure 8. As the concentration increases from 0.1 to 0.6 M, there is a slight decrease in pH. In this regard, Montoya et al. [43] have previously reported that the increase in the pH value of the electrolyte on top of AAs with increasing NaCl content from 0.06 to 0.6 M in static electrolytes is the result of a competition between $Cl^-$ and $OH^-$ ions in forming aluminum complex compounds through the process of Al hydrolysis. At the lowest concentration (0.01 M), Reactions (4), (5), and (6) are primarily promoted, leading to an acidic condition of the electrolyte (a pH around 4.0).

$$Al^{3+} + OH^- \leftrightarrow AlOH^{2+} \tag{4}$$

$$AlOH^{2+} + OH^- \leftrightarrow Al(OH)_2^+ \tag{5}$$

$$Al(OH)_2^+ + OH^- \leftrightarrow Al(OH)_3 \tag{6}$$

Meanwhile, the acidification process in NaCl electrolytes with concentrations from 0.01 to 0.1 M is explained by Reactions (7) and (8), indicating that, when the presence of $Cl^-$ ions competes with $OH^-$ ions in relation to the amount of electrolyte present, aluminum complex compounds are formed, subsequently avoiding a drop in pH. Furthermore, the hydrogen evolution process presented in diluted electrolytes (Figure 7) is accompanied by a significant local alkalinization near the cathodically active regions, as indicated by a previous study [69]. This alkalinization facilitates an increase in the pH value in AA7075.

$$Al^{3+} + Cl^- \leftrightarrow AlCl^{2+} \tag{7}$$

$$AlOH^{2+} + Cl^- \leftrightarrow AlOHCl^+ \tag{8}$$

On the other hand, increasing the concentration of NaCl from 0.1 to 0.3 and 0.6 M results in the electrolyte on top of AAs becoming more acidic. Chloride ions react with Al to form aluminum chloride ions ($AlCl^{2+}$), which act as an acid by donating $H^+$ in a catalytic process, according to Equations (7) and (9), as already reported by Guseva et al. [79] and Goley and Nguyen [80].

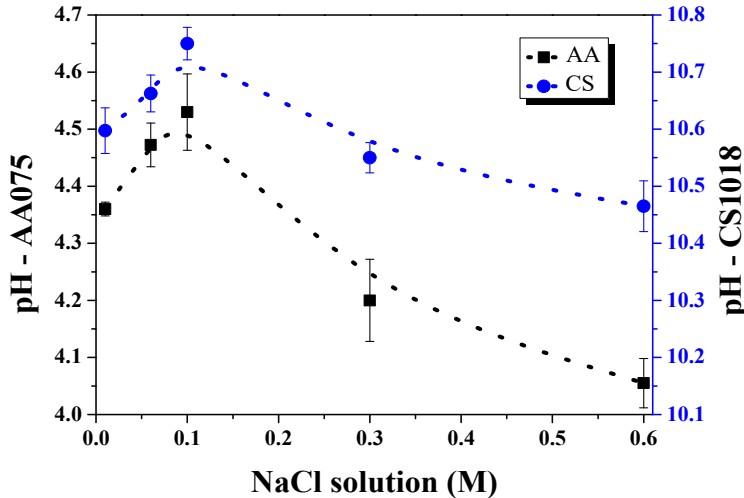

**Figure 8.** Evolution of average pH on the surface electrodes as a function of NaCl concentration after droplet evaporation.

It is important to note that, at a pH of approximately 4.0 in AAs, the primary reaction involving $H^+$ ions is the reduction of $H^+$ to $H_2$ gas, resulting in a higher pH. However, at higher NaCl concentrations and during the evaporation of the liquid electrolyte, there is an over-saturation of ions, affecting mass transportation and the distribution of chemical reactions, which contributes to pH changes, such as the $H^+$ evolution process [81]. Therefore, no significant evidence of $H_2$ gas is observed at a 0.6 M concentration (Figure 7), indicating an excess of $H^+$ ions and, as a result, a more acidic environment on the AA electrode.

$$AlCl^{2+} + 2H_2O \rightarrow Al(OH)_2Cl + 2H^+ \tag{9}$$

## 4. Conclusions

The galvanic corrosion behavior of the AA7075–CS1018 couple in dynamic electrolytes was studied using the ZRA technique. Based on the results of this study, the following conclusions were drawn.

The instantaneous magnitude of the galvanic current of the AA7075–CS1018 couple increases during droplet evaporation due to the presence of a more aggressive residual electrolyte. Nevertheless, this behavior cannot continue until complete evaporation is achieved, due to the limited solubility of oxygen and challenges associated with mass transport under such specific conditions. Indeed, in our experiment, such a residual and

concentrated electrolyte induced significant alterations in both vaporization enthalpy and viscosity, leading to a consequential slowdown in the evaporation rate.

Employing a thin gel electrolyte beneath the liquid droplet electrolyte facilitated, in our study, the monitoring of the pH evolution of the electrolyte on both electrodes, confirming the acidification of the electrolyte on AA7075 and the alkalization of the electrolyte on CS1018. The processes of acidification and alkalization were notably influenced by the initial concentration of the NaCl electrolyte, although this dependence did not exhibit a direct proportionality across all concentration ranges.

The average magnitude of the galvanic current in the AA7075–CS1018 couple is influenced by the initial NaCl concentration in the electrolyte. Notably, in initially dilute electrolytes, the current showed a direct proportionality to the concentration, while in initially concentrated electrolytes, this relationship demonstrated an inverse effect.

In low chloride concentrations, there was a significant potential drop, causing the electrodes to be closer to isolated corrosion potentials, allowing a pronounced occurrence of self-corrosion in CS1018. However, in more concentrated solutions, the electrodes adopted the coupled potential.

As a consequence of the electrochemical cell's design, the third electrode, intended as the reference electrode, remained inactive throughout our experimentation. This constraint restricted the range of potential measurements and prevented the application of a polarization test.

While both the gel and liquid electrolytes shared several identical initial conditions (such as electrical conductivity, pH, and NaCl concentration) and the syneresis of the gel ensured that the metallic surface remained consistently wet, specific differences persisted. Notably, the absence of observable solid aluminum corrosion products in the gel electrolyte film suggests that it does not promote such precipitation reactions.

**Author Contributions:** Conceptualization, R.M.; methodology, M.M.; software, M.M.; validation, M.M.; formal analysis, R.M. and M.M.; investigation, M.M.; resources, R.M. and J.G.; data curation, M.M.; writing—original draft preparation, M.M.; writing—review and editing, M.M. and R.M.; visualization, R.M.; supervision, R.M.; project administration, R.M. and J.G.; funding acquisition, R.M. and J.G. All authors have read and agreed to the published version of the manuscript.

**Funding:** UNAM-DGAPA-PAPIIT (IG100623) and UNAM-DGAPA-POSDOC programs.

**Data Availability Statement:** The data that support the findings of this study are available from the corresponding author upon reasonable request.

**Acknowledgments:** The authors express sincere gratitude for the financial support provided by the UNAM-DGAPA-PAPIIT program IG100623. Marvin, Montoya-Rangel gives thanks for his postdoctoral fellowship through the program POSDOC of DGAPA. Beyond the laboratory and lecture hall, Digby D. Macdonald has served as a mentor, guiding light, and role model for countless students and colleagues, instilling in them a legacy of curiosity, ethics, and an unwavering pursuit of knowledge. As Digby D. Macdonald transitions into retirement, we extend our deepest gratitude for his contributions to our scientific community and the indelible mark he has left on the wider world of science. Your work continues to inspire, and your footsteps will guide the paths of future scholars.

**Conflicts of Interest:** The authors declare no conflicts of interest.

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
