# Peer review of "The AA7075–CS1018 Galvanic Couple under Evaporating Droplets"

_cmd, doi:10.3390/cmd5010005_

Round 1
Reviewer 1 Report
Comments and Suggestions for Authors
The paper goes into a lot of detail on coupling currents and dependence on chloride concentration and evaporation. There is not too much wrong with it, although there are a few errors, but the data could have been explained better. In particular, there was not enough focus on the significance of potential drop at low concentrations of salt - indeed that was not explicitly mentioned.
With reference to NaCl concentration it should be emphasised that these are the nominal values but may change with evaporation.
p13, line 477. This sentence beginning Second, .... is nonsense. NaCl plus H2O does not produce HCl. This must be changed. The scientific description of the three factors here is poor.
Equations 9 and 10 are chemically unbalanced and must be changed.
In the second Conclusion the current is directly proportional to the concentration - not the other way round.
4th conclusion should simply be a statement that in low chloride there is significant potential drop so tendency is for electrodes to be closer to isolated corrosion potentials but in more concentrated solution the electrodes adopt the coupled potential.
Comments on the Quality of English Language
A few examples with sentence construction poor (eg. just before equation 3) but on the whole tolerable
Author Response
Authors.-First of all, the authors appreciate the time and effort that the Reviewer put in commenting on our work.
R1.-The paper goes into a lot of detail on coupling currents and dependence on chloride concentration and evaporation. There is not too much wrong with it, although there are a few errors, but the data could have been explained better. In particular, there was not enough focus on the significance of potential drop at low concentrations of salt - indeed that was not explicitly mentioned.
Authors.-Thanks for the feedback.
Due to the design of the electrochemical cell, measuring the potential evolution within the galvanic cell was not feasible, and therefore, it was not initially considered in the discussion. It should bear in mind that using a reference electrode in such a small configuration was not straightforward. However, it is well known that at low ion concentrations, there is an increase in electrical resistance leading to a subsequent increase in the potential drop, which prevents galvanic coupling. Therefore, in the new manuscript, reference [75] was added to support the self-corrosion observed in CS1018 (line 479) based on this phenomenon. Moreover, the 4th conclusion was modified according to potential drop as suggested also by the Reviewer.
R1.-With reference to NaCl concentration it should be emphasised that these are the nominal values but may change with evaporation.
Authors.-The authors consider that this information was already included in the original manuscript (see line 197, Figure 5, paragraph in lines 318-343). However, in the new manuscript now we include new lines emphasizing what the Reviewer requires (lines 251-245). Indeed, this time the abstract emphasized such information. Thank you for your comment.
R1.-p13, line 477. This sentence beginning Second, .... is nonsense. NaCl plus H2O does not produce HCl. This must be changed. The scientific description of the three factors here is poor.
Authors.-We apologize for the oversight, thank you for the comment. What the authors wanted to say is that NaCl electrolytes in the presence of aluminum generate protons as explained below.
As a strong substance, NaCl will dissolve completely in water. This means that it does not exist in their molecular form in aqueous solution; instead, it is all completely solvated into their constituent ions. Moreover, HCl won’t be formed because Na⁺ and Cl⁻ do not exhibit acidic or basic properties. However, the role of aluminum (Al) in the acidification of the electrolyte is based on the formation of aluminum chloride (AlCl3). When aluminum chloride comes into contact with water, it undergoes hydrolysis, resulting in the production of hydrochloric acid (HCl) and aluminum hydroxide (Al(OH)3) (Equation 6). The generated hydrochloric acid (HCl) increases the concentration of H+ ions in the solution, leading to a decrease in pH and making the solution more acidic. This information was briefly included in the manuscript in line 555.
R1.-Equations 9 and 10 are chemically unbalanced and must be changed.
Authors.-Thanks for the comment, the Equations have been balanced in the new version of the manuscript.
In the second Conclusion the current is directly proportional to the concentration - not the other way round.
Authors.-Indeed, the current is influenced by the initial concentration of NaCl. This confusion has already been rectified. Thanks for the comment.
R1.-4th conclusion should simply be a statement that in low chloride there is significant potential drop so tendency is for electrodes to be closer to isolated corrosion potentials but in more concentrated solution the electrodes adopt the coupled potential.
Authors.-Thank you for your comment. On line 479, information was added to support the conclusion suggested by you. Furthermore, the aforementioned conclusion was also modified.

Reviewer 2 Report
Comments and Suggestions for Authors
This paper was well prepared and organized from the point that it showed the galvanic corrosion between AA7075 and CS1018 under evapoating droplets. But some part of the manuscript needs to be revised as follows.
0. When the authors respond the answer, please revise the manuscript and upload the file including the below questions and answers.
1. The references about corrosion in microdroplet are more needed in Introduction.
2. Figure 1: Was AA7075 directly contacted to CS1018(Figure 1a)? If not, show the distance between them in Figue 1e. Was test solution present on the agar gel only? Or, was the droplet wetted on the two metal's surface in addition to the agar? Was the agar gel electroconducted or resistant? Was the gel porous? Those are not clear.
3. All instruments used need the detail information about 'model, manufacturer, country'.
4. line 162: Where is 'Discussion'?
5. Figure 3: Did you check and analyze the concentration of NaCl with evaporation time?
6. Figure 4; What is 'horas'?(1 mm means?) What are the numbers on the droplet? Unit? Real photos?
7. Figure 5: Describe the meaning of the arrows in the figure. Why didn't you show the evaporation of water(H2O)? The evaporation contents of NaCl and O2 depend upon the size of droplet? Please describe them.
8. Figure 6: Describe the working electrode in the title. I wonder what Al always was the anode in the dilute solution. In the dilute NaCl solution, Al could be the cathode. Describe the detail.
9. Why didn't you perform polarization test?
10. Table 2 needs the line.
11. Check the subscript etc.
12. Figure 7: Denote 'Anode' and Cathode'. Please check the polarity by the concentration of NaCl. Test solution was neutral, but this photos were the evidence of hydrogen gas evolution? Was it true?
13. Figure 8: What is the initial pH before test? How did you measure the pH of the drolpet? I think this system was the couple of 'corrode metal vs. corrrode metal(maybe, Al could be the passive state).
14. In equation, correct the arrows, NOT <->. Maybe -> is correct because of non-equibrium state.
15. Eq. (6), check AlOH^2^+ or AlOH^2+.
16. Eq.(9) and (10) were wrong. Revise them.
17. Based on the above points, the conclusions need to be deeper and clearer on also the drawbacks of this process, it may be fine to generally promote this process, but the authors should provide also a comprehensive and objective list of conclusions with the good the bad and the neutral conclusions.
Author Response
Reviewer 2
Authors.- First of all, the authors appreciate the time and effort that the Reviewer put in commenting on our work.
This paper was well prepared and organized from the point that it showed the galvanic corrosion between AA7075 and CS1018 under evaporating droplets. But some part of the manuscript needs to be revised as follows.
- When the authors respond the answer, please revise the manuscript, and upload the file including the below questions and answers.
- The references about corrosion in microdroplet are more needed in Introduction.
Authors.- Suggestion accepted, thanks. References 33 to 38 were inserted into the document in the new paragraphs included into the lines 79 and 97.
- Figure 1: Was AA7075 directly contacted to CS1018(Figure 1a)? If not, show the distance between them in Figue 1e.
Authors.- AA7075 and CS1018 plates were arranged horizontally side by side with a separation distance of 0.5 mm, as indicated in the new line 154: this information has been included in Figure 1e. Thank you for this observation.
Was test solution present on the agar gel only? Or, was the droplet wetted on the two metal's surface in addition to the agar?
Authors.- The liquid electrolyte was applied on top of the agar gel electrolyte, as illustrated in Figure 1c. Petroleum jelly (Vaseline) was used around the electrolytes to prevent any leakage or spillage of the liquid electrolyte onto the surfaces of the two metals.
However, it is essential for the reviewer to consider that the metallic surface remains wet at all times due to the property of syneresis in gel electrolytes. This property enables the continuous release of liquid from the gel to the solid surface where it is applied, ensuring that the surface stays wet.
Part of this information has been incorporated into lines 164 of the manuscript.
Was the agar gel electroconducted or resistant? Was the gel porous? Those are not clear.
Authors.- Regarding the agar gel electrolyte, and as previously reported, “Agar-based polymeric electrolytes have excelled technologically in the study of electrochemical phenomena [Ruiz-Garcia]” and (as written in the original manuscript) “the use of a gel electrolyte in this investigation is based on the fact that hydrogels, as agar and agarose, preserves the properties of diffusion coefficients and do not affect the reversible charge transfer on electrode reactions”. On the other hand, the gel electrolyte is considered an ion-conducting medium as it facilitates the movement of ions within its structure. As demonstrated in [Ruiz-Garcia], its conductivity can increase up to 100 times compared to when it is prepared with deionized water, particularly when prepared with NaCl solutions as in the present study [Ruiz-Garcia]. On the other hand, agar gels are generally known to possess some degree of porosity, which allows for the diffusion of ions and molecules. However, the extent of porosity can vary depending on factors such as agar concentration, gel preparation method, and environmental conditions. In the context of electrochemical experiments, such as ours, the agar gel served as an electrolyte matrix through which ions move, facilitating the electrochemical reactions described in the manuscript occurring at the electrode interfaces. Although not entirely porous like a conventional liquid electrolyte, the agar gel does demonstrate permeability to ions, rendering it suitable for electrochemical studies [Moon et al., Ahmad et al.]
Ruiz-Garcia, A.; Esquivel-Peña, V.; Genesca, J.; Montoya, R. Advances in Galvanic Corrosion of Aluminum Alloys. Electrochim Acta 2023, 449, 142227, doi:10.1016/J.ELECTACTA.2023.142227.
Moon, W.G.; Kim, G.P.; Lee, M.; Song, H.D.; Yi, J. A Biodegradable Gel Electrolyte for Use in High-Performance Flexible Supercapacitors. ACS Appl Mater Interfaces 2015, 7, 3503–3511, doi:10.1021/AM5070987/SUPPL_FILE/AM5070987_SI_001.PDF.
Ahmad Ruzaidi, D.A.; Mahat, M.M.; Mohamed Sofian, Z.; Nor Hashim, N.A.; Osman, H.; Nawawi, M.A.; Ramli, R.; Jantan, K.A.; Aizamddin, M.F.; Azman, H.H.; et al. Synthesis and Characterization of Porous, Electro-Conductive Chitosan–Gelatin–Agar-Based PEDOT: PSS Scaffolds for Potential Use in Tissue Engineering. Polymers 2021, Vol. 13, Page 2901 2021, 13, 2901, doi:10.3390/POLYM13172901.
- All instruments used need the detail information about 'model, manufacturer, country'.
Authors.- This information has been added in the experimental section. Thanks for the observation.
- line 162: Where is 'Discussion'?
Authors.- Thanks for the observation, the new version of the manuscript specified much better what we wanted to point out. (P.S. we believe that the line marked for the Reviewer is not 162 but 384).
- Figure 3: Did you check and analyze the concentration of NaCl with evaporation time?
Authors.- The concentration of NaCl was not directly monitored during the evaporation process. However, it is always possible to calculate it since we certainly measured both the transient droplet volume and the original salt content (which in principle does not vary due to its non-volatile nature). As indicated throughout the manuscript, it was consistently assumed that the concentration increased as the evaporation process was continuous. Therefore, while direct monitoring of NaCl concentration during evaporation was not conducted, this relationship was inferred based on established principles and observed trends.
Figure 4; What is 'horas'?(1 mm means?) What are the numbers on the droplet? Unit? Real photos?
Authors.- Thanks for this remark, horas is hours in Spanish. We have amended such a mistake.
On the other hand, each photo corresponds to a specific time in hours (h) or seconds (s) of the real experimental setup. The figure has been modified based on your feedback.
Regarding the photos, the figure caption has been updated to provide a clear explanation of the changes.
- Figure 5: Describe the meaning of the arrows in the figure.
Authors.- The larger the number of upward arrows the higher the concentration of both NaCl and O2. new lines have been added to the new version of the manuscript (lines 324-328). Also, the caption of Figure 5 has been updated to include such information.
Why didn't you show the evaporation of water(H2O)? The evaporation contents of NaCl and O2 depend upon the size of droplet? Please describe them.
Authors.- In Figure 5, the evaporation of H2O was schematically depicted as a reduction in the volume (weight loss) of the liquid electrolyte. As shown in Figure 3, different evaporation rates are present depending on the initial NaCl concentration and the accumulation of NaCl through evaporation. It should be considered that also in Fig 3 the real transient loss of water was provided. Also please take into account the answer provided to your question number 5.
Between lines 197 and 245, various factors influencing evaporation are discussed. Additionally, from lines 235 to 245, further information was added to complete this discussion, highlighting how a NaCl concentration hinders evaporation by impeding the diffusion of water molecules.
- Figure 6: Describe the working electrode in the title. I wonder what Al always was the anode in the dilute solution. In the dilute NaCl solution, Al could be the cathode. Describe the detail.
Authors.- Certainly, the aluminum alloy was always the working electrode as described in Figure 2 and such information was also included in the new version of the manuscript as caption of Figure 6. On the other hand, the very interesting possibility that the Reviewer suggests (that the aluminum alloy could work as cathode) was dismissed since the resulting current was always positive. Indeed some of our preliminary results in this regard show that once the CS1018 is taken as the working electrode the current is always negative (See figure below).
Authors.- Indeed, both electrodes exhibited anodic and cathodic components, as explained from line 462. However, the AA7075 consistently functioned as the anode within the galvanic couple, as evidenced by the absence of a change in the sign of the galvanic current. This behavior is explained in the new manuscript as follows:
“In low chloride concentrations, there is a significant potential drop, causing the electrodes to be closer to isolated corrosion potentials, allowing a pronounced occurrence of self-corrosion in CS1018. However, in more concentrated solutions, the electrodes adopt the coupled potential.”
- Why didn't you perform polarization test?
Authors.- Our intention was to determine the corrosion behavior under atmospheric conditions, with a focus on monitoring the natural evolution (without polarization) of the galvanic couple. To accomplish this, it was necessary to establish the open-circuit condition and utilize the ZRA technique to measure the current evolution. Moreover, due to the design of the electrochemical cell, it is difficult to add a reference electrode to sense the potential. This limitation was included in the new conclusion section.
- Table 2 needs the line.
Authors.- Although the suggestion of the Reviewer is not completely clear for the authors, the new version of the manuscript includes a horizontal line in both Table I and Table II.
- Check the subscript etc.
Authors.- We have double checked the document. Thank you for the suggestion.
- Figure 7: Denote 'Anode' and Cathode'. Please check the polarity by the concentration of NaCl. Test solution was neutral, but this photos were the evidence of hydrogen gas evolution? Was it true?
Authors.- According to standard potential, the 'anode' is the AA7075, and CS1018 is the cathode. This was evident in the ZRA measurement results reported on line 262, where we noted that the positive current density values registered indicate AA7075 corrosion in all cases, with CS1018 acting as the cathode. However, as the initial NaCl concentration decreased, as shown in Figure 7, a significant potential drop occurred, resulting in an almost isolated corrosion potential between both electrodes. In more concentrated solutions, the electrodes adopted a coupled potential. Therefore, both electrodes exhibited anodic and cathodic components. However, the AA7075 consistently behaved as an global anode within the galvanic couple.
This helps explainings the occurrence of self-corrosion of CS.
The authors believe that labeling each electrode as "Anode" and "Cathode" may cause confusion among readers, particularly considering the self-corrosion behavior of steel.
On the other hand, despite the initial pH of the electrolyte being neutral, the dissolution of aluminum resulted in its ions hydrolysis, causing a decrease in electrolyte pH to around 4.3, as depicted in Figure 8. The bubbles observed may be attributed to: i) the reduction of H⁺ to produce H₂ due to the increase in cathodic current in AA7075 caused by the self-corrosion of CS1018, as explained in the manuscript, and/or ii) the negative difference effect present in aluminum and magnesium alloys which is able to reduce the water. This information was added on line 435 of the manuscript.
- Figure 8: What is the initial pH before test? How did you measure the pH of the drolpet? I think this system was the couple of 'corrode metal vs. corrrode metal(maybe, Al could be the passive state).
Authors.- The initial pH of both phases was adjusted to 7.0 using dilute NaOH solution (as mentioned in line 140). The final pH values were determined from the chemical information left by the electrolyte droplet, once it was evaporated, on the agar electrolyte. Indeed, once the complete evaporation of the liquid electrolyte took place, and prior to the evaporation of moisture from the agar gel electrolyte, the gel was removed from the galvanic couple, and the pH of such gel was measured (both above the AA and the CS). This information was added in line 154.
On the other hand, as you suggest, both electrodes experienced corrosion due to an imbalance in the cathodic and anodic currents of each electrode caused by variations in NaCl concentration. And this has been explained in terms of a high potential drop in dilute electrolytes (lines 595).
However, the possibility that Al alloy could have behaved as a cathode (passive state) is not supported by the ZRA measurements that always provided a positive current using the experimental configuration of Fig 2.
- In equation, correct the arrows, NOT <->. Maybe -> is correct because of non-equibrium state.
Authors.- Thanks for the observation, we have corrected the corresponding arrows of Reaction 2 and 3. Regarding the homogeneous reactions between chloride, Al3+ ions, and Al hydrolysis different equations were verified.
Guseva, O.; Derose, J.A.; Schmutz, P. Modelling the Early Stage Time Dependence of Localised Corrosion in Aluminium Alloys. Electrochim Acta 2013, 88, 821–831, doi:10.1016/J.ELECTACTA.2012.10.059.
- Eq. (6), check AlOH^2^+ or AlOH^2+.
Authors.- Thanks for the observation. However, authors have reviewed the reactions of aluminum in water in presence of Cl and confirmed that compounds in Equation 6 are correctly written. This information can be documented in the following references.
Al(OH)2++ OH-↔Al(OH)3
Montoya, R.; Ruiz-García, A.G.; Ortiz-Ozuna, A.; Ramírez-Barat, B.; Genesca, J. Acidification of the Electrolyte during the Galvanic Corrosion of AA7075: A Numerical and Experimental Study. Materials and Corrosion 2021, 72, 1259–1269, doi:10.1002/MACO.202012274.
Abodi, L.C.; Gonzalez-Garcia, Y.; Dolgikh, O.; Dan, C.; Deconinck, D.; Mol, J.M.C.; Terryn, H.; Deconinck, J. Simulated and Measured Response of Oxygen SECM-Measurements in Presence of a Corrosion Process. Electrochim Acta 2014, 146, 556–563, doi:10.1016/J.ELECTACTA.2014.09.010.
Foley, R.T.; Nguyen, T.H. The Chemical Nature of Aluminum Corrosion: V . Energy Transfer in Aluminum Dissolution. Proceedings - The Electrochemical Society 1981, 81–8, 27–36, doi:10.1149/1.2123881/XML.
Martins, J.I.; Diblikova, L.; Bazzaouia, M.; Nunesa, M.C. Polypyrrole Coating Doped with Dihydrogenophosphate Ion to Protect Aluminium against Corrosion in Sodium Chloride Medium. J Braz Chem Soc 2012, 23, 377–384, doi:10.1590/S0103-50532012000300002.
Guseva, O.; Derose, J.A.; Schmutz, P. Modelling the Early Stage Time Dependence of Localised Corrosion in Aluminium Alloys. Electrochim Acta 2013, 88, 821–831, doi:10.1016/J.ELECTACTA.2012.10.059.
- Eq.(9) and (10) were wrong. Revise them.
Authors.- Thanks for this remark. The equations were balanced and modified according to the homogeneous reactions between chloride, Al3+ ions, and the Al hydrolysis products reported by Guseva.
Guseva, O.; Derose, J.A.; Schmutz, P. Modelling the Early Stage Time Dependence of Localised Corrosion in Aluminium Alloys. Electrochim Acta 2013, 88, 821–831, doi:10.1016/J.ELECTACTA.2012.10.059.
- Based on the above points, the conclusions need to be deeper and clearer on also the drawbacks of this process, it may be fine to generally promote this process, but the authors should provide also a comprehensive and objective list of conclusions with the good the bad and the neutral conclusions.
Authors.- The authors accept your suggestion.

Round 2
Reviewer 2 Report
Comments and Suggestions for Authors
1. Check '3. Results' and '5.Conclusions'. Where is 4. Discussion?
2. Table 3 needs the horizontal line under NaCl concentrations.